# Stable Local Bit-Line 6 T SRAM Architecture Design for Low-Voltage Operation and Access Enhancement

**Ming-Hwa Sheu [1], S M Salahuddin Morsalin [1,*], Chang-Ming Tsai [1], Cheng-Jie Yang [1], Shih-Chang Hsia [1], Ya-Hsin Hsueh [1], Jin-Fa Lin [2] and Chuan-Yu Chang [3]**

1. Department of Electronic Engineering, National Yunlin University of Science and Technology, Douliu City, Yunlin County 64002, Taiwan; sheumh@yuntech.edu.tw (M.-H.S.); m10813014@gmail.yuntech.edu.tw (C.-M.T.); M10613211@yuntech.edu.tw (C.-J.Y.); hsia@yuntech.edu.tw (S.-C.H.); hsuehyh@yuntech.edu.tw (Y.-H.H.)
2. Department of Information and Communication Engineering, Chaoyang University of Technology, Wufeng District, Taichung City, Taichung 413310, Taiwan; jflin@cyut.edu.tw
3. Computer Science and Information Engineering, National Yunlin University of Science and Technology, Douliu City, Yunlin County 64002, Taiwan; chuanyu@yuntech.edu.tw
* Correspondence: s.morsalin10@gmail.com

**Abstract:** To incur the memory interface and faster access of static RAM for near-threshold operation, a stable local bit-line static random-access memory (SRAM) architecture has been proposed along with the low-voltage pre-charged and negative local bit-line (NLBL) scheme. In addition to the low-voltage pre-charged and NLBL scheme being operated by the write bit-line column to work out for the write half-select condition. The proposed local bit-line SRAM design reduces variations and enhances the read stability, the write capacity, prevents the bit-line leakage current, and the designed pre-charged circuit has achieved an optimal pre-charge voltage during the near-threshold operation. Compared to the conventional 6 T SRAM design, the optimal pre-charge voltage has been improved up to 15% for the read static noise margin (RSNM) and the write delay enriched up to 22% for the proposed NLBL SRAM design which is energy-efficient. At 400 mV supply voltage and 25 MHz operating frequency, the read and write energy consumption is 0.22 pJ and 0.23 pJ respectively. After comparing with the related works, the access average energy (AAE) is lower than in other works. The overall performance for the proposed local bit-line SRAM has achieved the highest figure of merit (FoM). The designed architecture has been implemented based on the 1-Kb SRAM macros and TSMC−40 nm GP process technology.

**Keywords:** energy-efficient; local bit-line; static random-access memory; near-threshold operation; low-voltage pre-charged; negative local bit-line; read static noise margin

## 1. Introduction

Modern electronics are being merged into smart technologies such as the Internet of Things (IoT), automotive electronics, biomedical electronics, sensor devices, and so on. Therefore, integrated circuits are widely used for low power consumption, low leakage current, and compact area [1]. However, with the development of nanometer (nm) process technology, the effect of leakage current has become a major problem in the system on chip (SoC) [2]. Modern microprocessors require more embedded memory for system specification, compact area, low energy, and power consumption [3,4]. The embedded memory consumes most of the power, and the large memory is used for data storage in the SoC. Thus, every circuit designer must pay attention to reduce power consumption. Conversely, the hostile design and size constraints make it much more difficult than general logic circuits to minimize the operating voltage [5–7]. As a result, several designers have connected additional circuits to decrease power consumption and increase operating stability for low supply voltages [8].

The conventional (conv.) 6 T SRAM performance is ineffective at the low supply voltage because of pseudo-read error in half-select condition. A read-decoupled 8 T SRAM [9,10] architecture was presented to solve the read error although minimized read error and increased RSNM slightly. Conversely, the memory cell is still affected by the read error during the write operation. Adding stacked access transistors, several SRAMs architectures the 9 T SRAM [11–13], 10 T SRAM [14,15], and 12 T SRAM [16] for near-threshold/ sub-threshold operations have been proposed to address read errors in half-select conditions. As a result of stacked access transistors, the write operation becomes weak. The bit-line of SRAM is deeply affected by large parasitic capacitance during read and write operations. The Average−8 T SRAM [17] consisted of local and global bit-lines proposed to solve the parasitic problem. However, the local and global bit-lines cannot achieve full swing due to less write ability, slow operation, and high-power consumption. The full-swing local bit-line SRAM architecture [18] was proposed but still poor write ability because two cascade transistors controlled the bit cell. The 10 T SRAM [19] stack pull-down transistors for a cross-coupled inverter with VGND technology was proposed to change the write path, but the write ability is still inefficient.

After analyzing the relationship between read noise and pre-charge voltage of the bit-line pair, a new local bit-line 6 T SRAM architecture has been proposed to improve the noise margin of SRAM, capacitive density, read stability, as well as write capacity.

## 2. Proposed Local Bit-Line SRAM Architecture

A modern 6 T SRAM architecture has been proposed to robust the performance, the block diagram shown in Figure 1. Consisting with four bits of 6 T cells, two assistant circuits for optimal pre-charge (OPC), and the NLBL framework have been developed for the proposed local bit-line SRAM architecture. The OPC circuit is control by the block selection lines of BLK [0] and BLKB [0] which are generated by row address decoder. The global read bit-line pair (GRBL [0] and GRBLB [0]) reads data from LBL [00] and LBLB [00] via the two transistors T1 and T2. The global write bit-line pair (GWBL [0] and GWBLB [0]) writes strong data in the NLBL and passes to the local bit-line pair (LBLB [00], LBL [00]) and SRAM cells for operation.

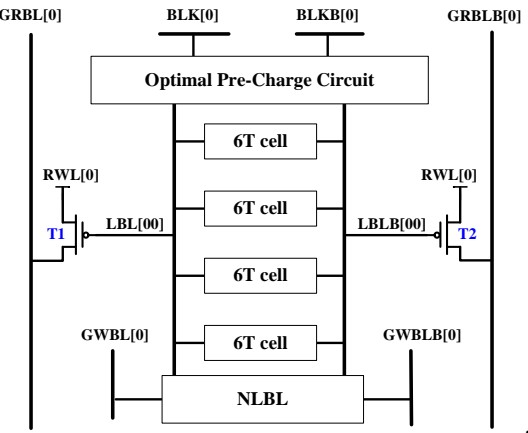

**Figure 1.** The proposed local bit-line 6 T SRAM block diagram.

### 2.1. Optimal Pre-Charge Circuit and Read Operation

To facilitate the read operation, an OPC circuit relates to the proposed SRAM design shown in Figure 2. The OPC circuit is controlled by the transistors LPL, LPR, NCL, NCR, and EQ. Initially, the block selection line set BLKB [0] = 0 and BLK [0] = 1 to turn off the pre-charge circuit, when BLKB [0] = 1, the optimal voltages ($V_{opt}$) are obtained by the transistors LPL and LPR using the following equation:

$$V_{opt} = V_{BLKB[0]} - Ids \times Rds \tag{1}$$

where '***Ids***' stands for the saturation current, and '***Rds***' channel resistance in LPR or LPL transistors and '***V<sub>BLKB</sub>***'is the voltage for block line BLKB[0].

The pre-charge voltage for near-threshold operation and process variance is not equal to the local bit-line LBL [00] and LBLB [00]. The block line BLK [0] = 0 is used to facilitate LBL [00] and LBLB [00] for voltage equalization by the transistor EQ. Table 1 shows the OPC voltages for different supply voltages.

**Table 1.** Optimal pre- charge voltages.

| Supply Voltage for Operation | Optimal Voltage Range (mV) |
| --- | --- |
| 0.9 V | 600~650 |
| 0.8 V | 550~600 |
| 0.7 V | 500~550 |
| 0.6 V | 400~450 |
| 0.5 V | 350~400 |

Four memory cells connected with local bit-line pair LBL [00] and LBLB [00] which minimizes the parasitic capacitance on the bit-line and reduces the read error. The memory cell reads data "1" and the local bit-line LBLB [00] is discharged into the cell through the transistor T4. The transistor T3 and transistor NCL have a small current, the four memory cells of the local bit-line significantly decrease the leakage currents. To increase the read stability and improve the RSNM, the global read bit-line (GRBL/GRBLB) avoids memory cell leakage currents. The sense amplifier (SA) reads the data out to charge the global read bit-line GRBLB [0] through transistor T2, there has a small parasitic capacitance which makes the read operation faster. The voltage of block selection lines BLKB [0] and BLK [0] is not less than V<sub>opt</sub>, so the low-voltage pre-charge circuit saves optimal voltage (V<sub>opt</sub>) and ensures read stability of read operation.

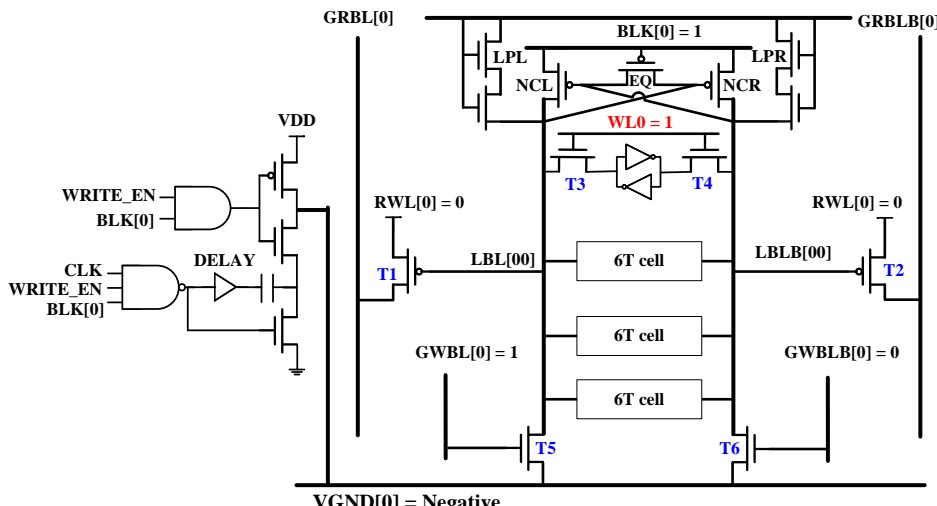

**Figure 2.** Proposed design of local bit-line 6 T SRAM Architecture.

*2.2. Negative Local Bit-Line Scheme and Write Operation*

Figure 2 displays the negative local bit-line scheme and the process of the write operation. The global write bit-line pairs GWBL [0] and GWBLB [0] are operated for write data by the following transistors T5 and T6. The Negative Virtual Ground (NVGND) in the column direction is attached to the LBL [00] or LBLB [00] to switch on the transistor T5 or T6, then pulls the other side to VDD via cross transistors NCL and NCR. At first, the block selection lines set BLKB [0] is '0', BLK [0] as '1' then the GWBL [0] or GWBLB [0] set '1', to switched on transistor T5 and T6. The other side of the bit-line LBL/ LBLB is engaged through the NCL/NCR to the VDD, and the word-line WL [0] sets '1' to write for the

memory cell. By sharing the charging capacitor, memory cells are discharged via VGND to improve the write capabilities for the near-threshold operation. For instance, the first cell discharge path through T3 to LBL [00] to pass data '0', and then the LBLB [00] retained the VDD to provide additional data. In this process the LBL [00] is connected to the NLBL, the memory cell data is reversed faster during the write operation and RWL [0] is still holding data '0'. For the differential read-write function the proposed local bit-line SRAM has high speed, decreased parasitic capacitance, and resistance which is energy-efficient.

### 2.3. Half-Select Condition Operation

Figure 3 illustrates the half-select condition operation for row blocks of proposed local bit-line SRAM. At the write operation of BLOCK 0, the transistors T3 and T4 are used to pass data of Q and QB which is indicated by the sky-blue color. Initially, the word-line WL0 =1 becomes active that shown by the red color and Q save data '1' and QB saves '0'. The global write bit-lines GWBL [0] =1 and GWBLB [0] =0 select data to activate the transistors T5 and T6 then the local bit-line LBL [00] turn out to be discharged through the transistor T5 to the negative voltage of VGND [0]. After discharge the local bit-line LBL [00], the Q = 0 and QB = 1 flip data in the memory cell. At BLOCK 1, the half-selected cell is affected by the word-line WL0 in the same row pseudo-read of LBL [10] and LBLB [10] that indicated by the orange color. The read word-line RWL [0] remains data "0" and row half-selected blocks does not charge by the global read bit-lines (GRBL/GRBLB), so the low-voltage pre-charge scheme reduces the read disturb and flip data in the memory cell. The proposed structure injects less pre-charge into cell and reduces the capacitance in the local bit-line which decline the leakage power consumption that shown by the purple color dotted line.

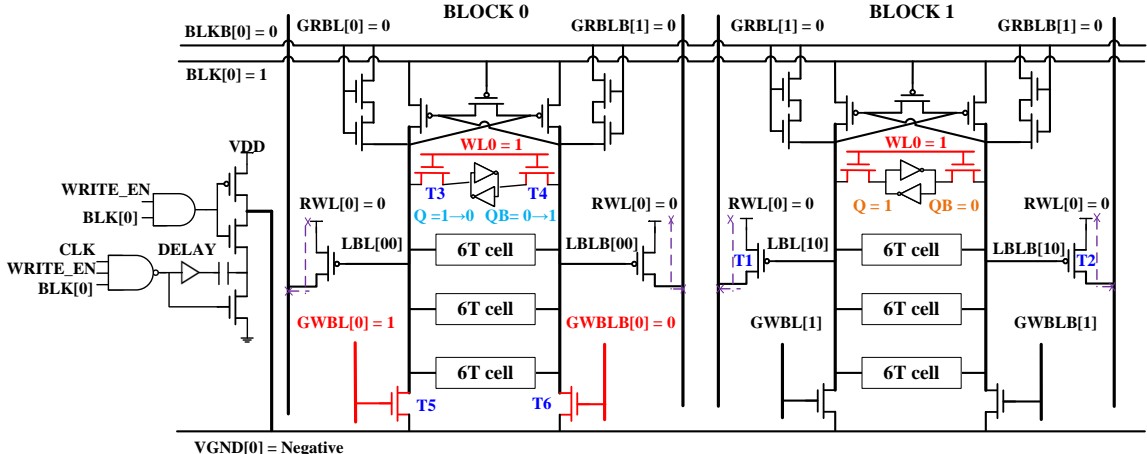

**Figure 3.** Half-Select Condition operation of local bit-line SRAM.

## 3. Simulation Results

### 3.1. The Comparison of RSNM Simulation

Figure 4 shows the RSNM simulation and analysis results for 6 T SRAM cell depends on the bit-line pre-charge voltage adjustment with different supply voltages. It is obvious that each curve has the maximum RSNM value at the specific bit-line voltage corresponding to the supply voltage. This simulation result is considered for proposed local bit-line design.

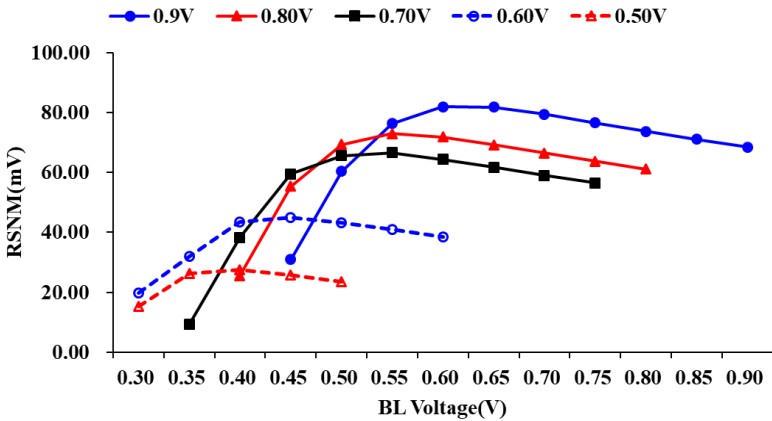

**Figure 4.** The RSNM simulation results of 6 T SRAM for pre-charge voltage adjustment.

The comparison of RSNM simulation result of proposed local bit-line SRAM and conv. 6 T SRAM at the various operating voltages is shown in Figure 5. The RSNM memory cell curve decreased for low voltages. The proposed local bit-line SRAM has obtained strong RSNM by using low-voltage pre-charged scheme during the read operation.

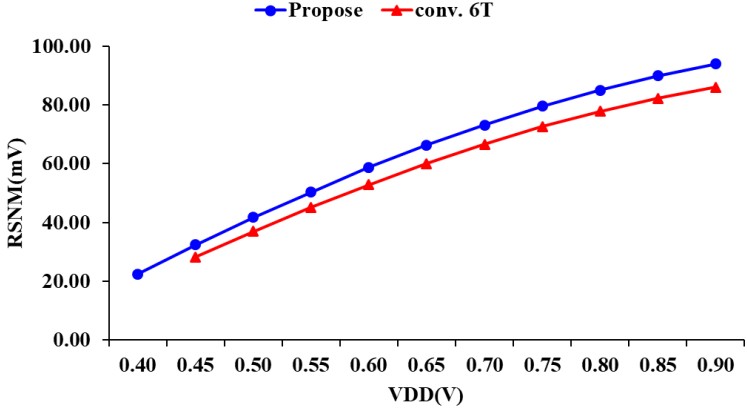

**Figure 5.** The RSNM comparison for optimal and (conv.) pre-charge.

### 3.2. The Comparison of Monte Carlo Simulation

For the global variations, the process-voltage-temperature (PVT) corners that combined the extreme cases of these variables are commonly used to verify the performance. The 10,000 times Monte Carlo post-simulations were performed for the experiment to show the enhancement of read stability. At 400 mV supply voltages, the experimental result of the conventional 6 T SRAM cell is shown in Figure 6a. During the read operation, the conv. 6 T SRAM charge on the bit-line pair destroyed the data store in the memory cell, flipped the data store in the cell, and reversed several read errors and data flips. Alternatively, the proposed local bit-line SRAM uses a low-voltage pre-charge scheme to optimize the voltage and achieved the stability of the cell without any read errors at the same supply voltage the result is shown in Figure 6b.

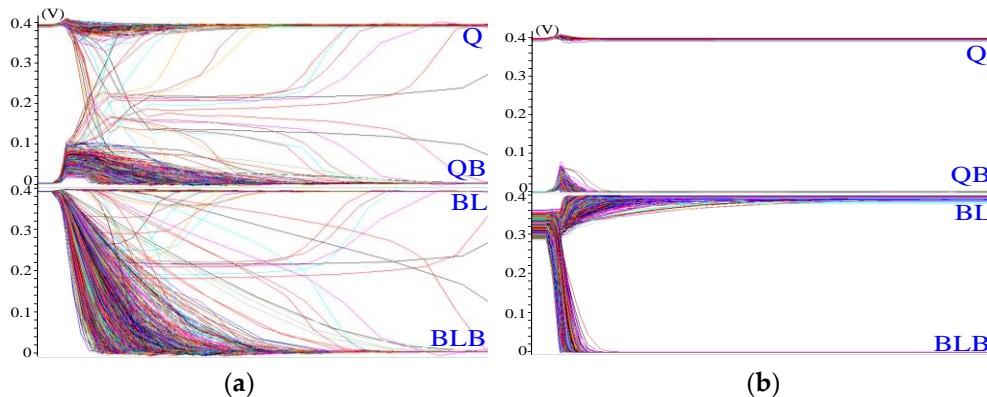

**Figure 6.** 10,000 times read Monte Carlo post-simulations at 400 mV (**a**) Result of (conv.) 6 T SRAM. (**b**) Result of proposed local bit-line SRAM.

### 3.3. The Comparison of Write Ability Simulation

The comparison and write ability simulation result for the different operating voltages is shown in Figure 7. The differential write operation is considered to write speed of data "0" and "1", the write operation is affected by the transition point of the cross-coupled inverter. However, the NLBL technology that included with the proposed design, so the write ability of the memory cells has improved and provides better performance. Compared to the conv. 6 T SRAM, the write speed of the proposed design has improved about 22% at the 400 mV operating voltage.

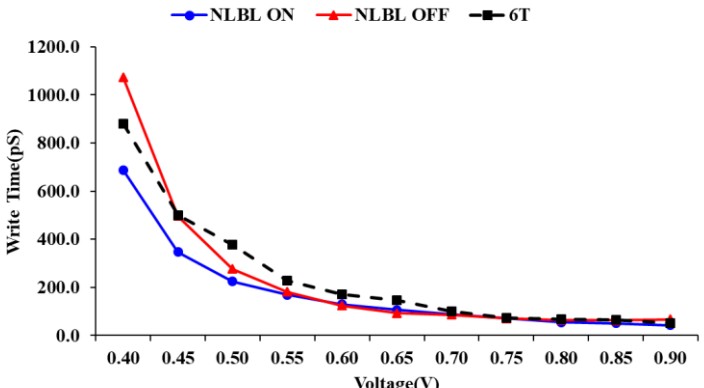

**Figure 7.** Write ability comparison at different operating voltages.

### 3.4. The Comparison of Bit-Line Swing Simulation

Figure 8 shows the difference between conv. 6 T SRAM and the proposed bit-line swing at 400 mV supply voltage. During the read operation, the bit-line pair has a different voltage because of the leakage current of all column half-select cells. The voltage difference of conv. 6 T SRAM is less than 200 mV at 128 bits, which causes the bit-line swing is too small for the SA that cannot succeed stability. The proposed local bit-line SRAM maintains the voltage difference of more than 300 mV and bit-line depth is changed to 256 bits.

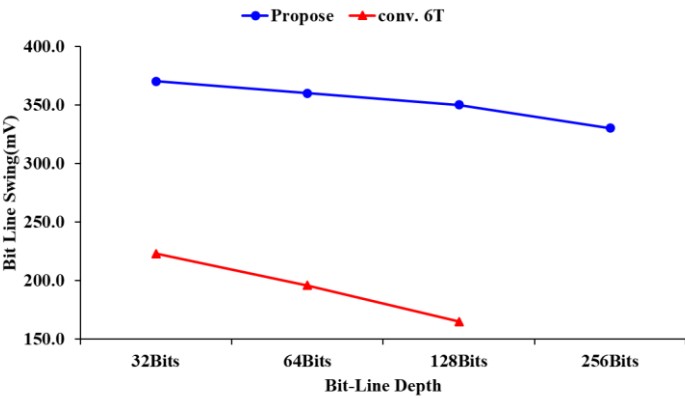

**Figure 8.** Comparison of bit-line swing at different bit-line depth.

### 3.5. The Comparison of Leakage Power

The static leakage power consumption of the memory cell is shown in Figure 9. Although the proposed architecture increases the metal–oxide–semiconductor field-effect transistor (MOSFET) number but effectively reduces the leakage current on the bit-line and the static leakage power consumption. The proposed design effectively reduces the bit-line leakage current and static leakage power consumption compared to the conv. 6 T SRAM.

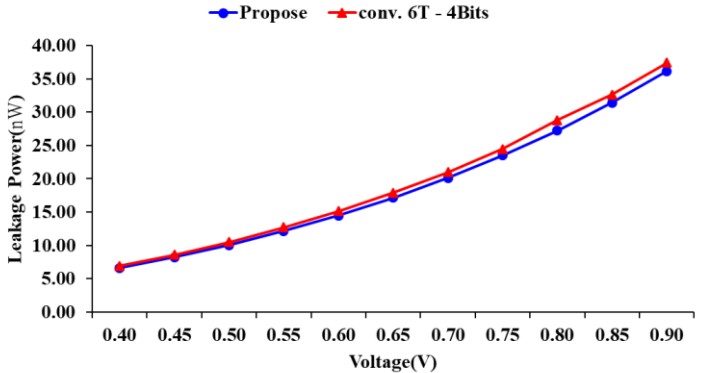

**Figure 9.** Comparison of leakage power at different operating voltages.

### 4. Chip Implementation and Result Comparison

The read and write operations are considered to construct the proposed local bit-line 6 T SRAM. While doing the operation, the read and write cell cannot destroy the logic in the cell. The uniqueness of the layout area, the cell size of the SRAM system has a great impact on the proposed design. The comparison of memory cell size and area is shown in Table 2. The 1 kb conv. 6 T area is still smaller than 8 T and proposed 6 T whereas, the proposed 6 T SRAM cell has a better performance for the read and write operation. The proposed local bit-line SRAM area is 7.65 $\mu m^2$ as shown in Figure 10a, and the light layout of the proposed design using the TSMC−40 nmGP process technology is shown in Figure 10b. Comparison table of the SRAM area.

**Table 2.** Comparison table of the SRAM area.

| Layout | Conventional 6 T | 8 T [10] | Proposed 6 T |
|---|---|---|---|
| 1 Kb area($\mu m^2$) | 1935.36 | 2466.20 | 2436.57 |
| size | (1×) | (1.27×) | (1.26×) |

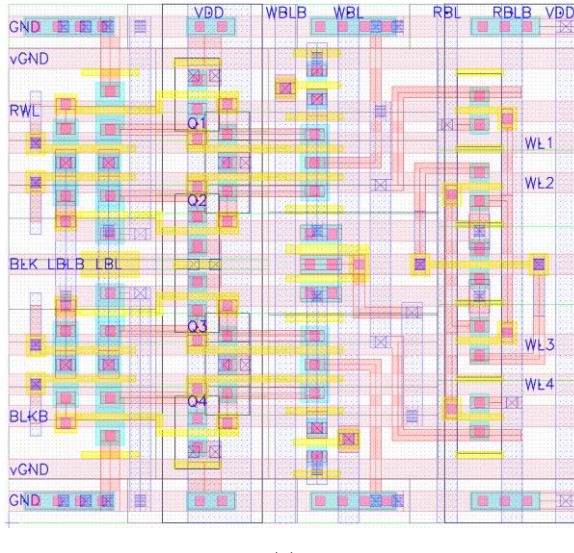

(**a**)

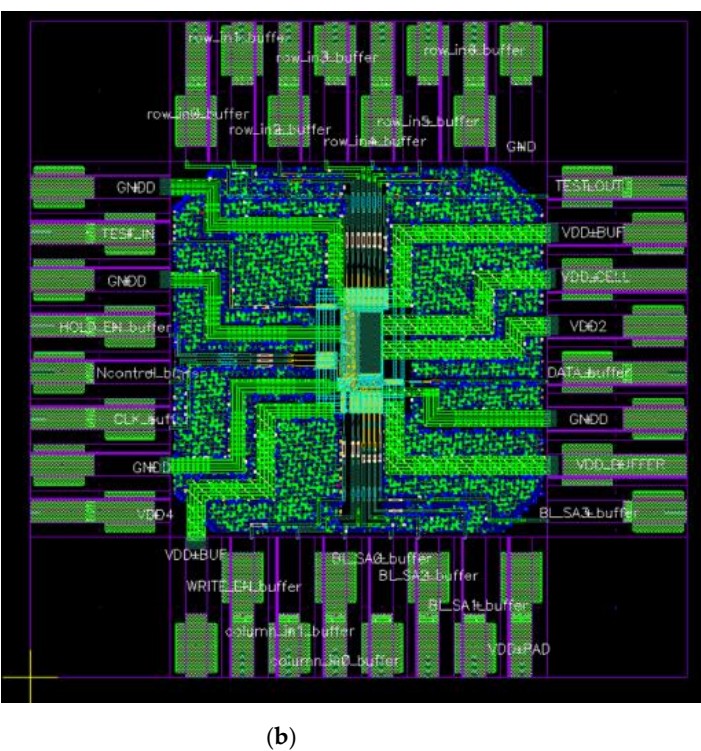

(**b**)

**Figure 10.** (**a**) Proposed 4-bit block area (**b**) Proposed 1 kb SRAM chip layout.

The implemented architecture 1 kb SRAM macros (128 rows × 8 columns) has 32 blocks in each column and each block consists of 4 memory cells. By using the Hspice EDA tool, the waveform of post-layout simulation is shown in Figure 11 at 400 mV/25 MHz. The WRITE_EN is the write control signal that controls the GWBL/GWBLB decoder, write driver circuit and NLBL scheme circuits to start work when WRITE_EN is "1". At first set input data is "1" to test the write status parameter. The selected memory cell stores data "1" during the write cycle "1" and reads the status, then the sense amplifier starts operating and pulls up the SA and latch to "1" which means the read and write operation both are successful. If WRITE_EN is "0" then the read operation starts to work. Similarly, when the write data is "0" the selected memory cell stored data "0" and the read status start, the SA and latch is also pulled down to "0". These parameters speed up the proposed local bit-line

SRAM to operate at the near-threshold operation by reducing read error strengthening the write ability.

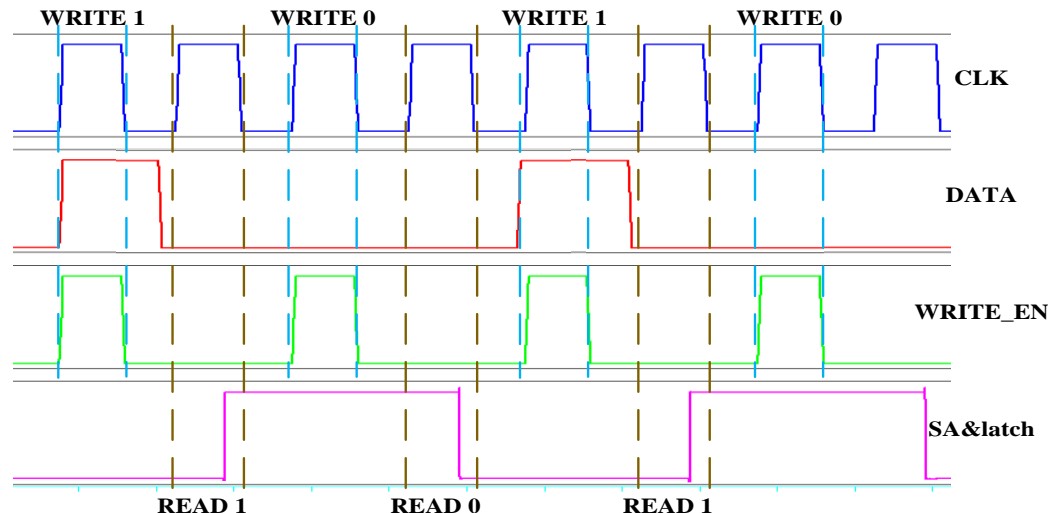

**Figure 11.** Post-layout simulation waveform for SRAM Chip @400 mV/25 MHz/TT corner.

Table 3 shows the comparison of proposed local bit-line SRAMs architecture performance and previous work. The comprehensive summary of the proposed local bit-line SRAM has smaller average energy consumption than the MINI-Array [20] and the highest FoM performance.

**Table 3.** Comparison of previous and proposed SRAM based on the local bit-line structure.

| Design Approach | Averge−8 T [17] | Full-Swing 6 T [18] | Pre-Charge 6 T [21] | Mini-Area 6 T [20] | Proposed |
|---|---|---|---|---|---|
| Technology | 130 nm | 22 nm | 22 nm | 28 nm HkMG | 40 nmGP |
| Area per block | 24.65 μm² | 0.71 μm² | 0.70 μm² | 1.36 μm² | 7.65 μm² |
| Local Bit-Line | Pre-Discharged | Pre-Discharged | Pre-Charged | Pre-Charged | Low-Voltage Pre-Charged |
| Write mechanism | Single-ended | Differential | Differential | Differential | Differential + NLBL |
| VDDMIN | 260 mV [*1] | 440 mV | 400 mV | 500 mV | 400 mV |
| Capacity | 64-kb | N. A | N. A | 256-kb | 1-kb |
| Area(mm²) | 0.512 | N. A | N. A | 0.397 | 0.00690 |
| Frequency @ VDDMIN | 245 kHz | N. A | N. A | 20 MHz | 25 MHz |
| Read energy (pJ) | 0.23 | 0.19 | 0.14 | N. A | 0.22 |
| Write energy (pJ) | 0.24 | 0.45 | 0.42 | N. A | 0.23 |
| Average Energy (pJ) | **0.23** | 0.32 | 0.28 | 0.468 | 0.22 |
| Leakage Power @VDDMIN | 0.88μW | N. A | N. A | 60μW [*2] | 3.363μW |
| FoM [*3] | 1× | 18× | 23.5× | 4.25× | 25× |
| Sim. /Meas. | Measured | Post-sim | Post-sim | Measured | Post-sim |

[*1] Size drastically adjustment required. [*2] Ultra-Low-Leakage Technology (HkMG CMOS or FinFET/Tri-Gate) and Power-gating scheme.
[*3] FoM=SNM/ (PDP² × normalized 1 block area) [22] is normalized to proposed local lit-line SRAM.

## 5. Conclusions

The low-voltage pre-charged and NLBL scheme have been included with the proposed local bit-line 6 T SRAM architecture which can be operated at the near-threshold operation. The low-voltage pre-charged circuit has reduced the read error and improved the read stability and RSNM of the memory cells. Moreover, the NLBL scheme has reduced the write error and improved the write ability for the near-threshold operation. Furthermore, the half-select cells pseudo-read error has reduced at the half-select condition in the proposed design. Likewise, the proposed architecture of the local bit-line SRAM eliminates the bit-line leakage induced and read failures. The TSMC−40 nmGP process technology has been implemented for the proposed local bit-line 6 T SRAM on 1 kb SRAM macros fabricated. At 400 mV supply voltage and 25 MHz operating frequency, write energy consumption

is saved about 45.2%, and the average energy consumption is reduced by about 52.9% compared to the MINI-Array. The proposed local bit-line 6 T SRAM effectively applicable to operate at low-power SoC chips.

**Author Contributions:** M.-H.S. and J.-F.L. proposed the idea and method; C.-M.T. and C.-J.Y. performed the simulations and experiments; S.-C.H. and C.-Y.C. analyzed the data; S.M.S.M. wrote the manuscript; M.-H.S. and Y.-H.H. reviewed the manuscript. All authors have read and agreed to publish this version of the manuscript.

**Funding:** This research is supported by the Ministry of Science & Technology, Taiwan under contract No. 109−2221-E−224 −050 and No. 109-2221-E-324-028.

**Data Availability Statement:** Required data is contained within the article.

**Acknowledgments:** The authors would like to acknowledge technical support for simulation by Taiwan Semiconductor Research Institute, Hspice EDA tool support for IC implementation.

**Conflicts of Interest:** The authors declare that no conflict of interest.

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
