# Peer review of "Stable Local Bit-Line 6 T SRAM Architecture Design for Low-Voltage Operation and Access Enhancement"

_electronics, doi:10.3390/electronics10060685_

Round 1

Reviewer 1 Report

Summary

First of all, get help from a native speaker to correct many English mistakes. As it is the paper cannot be published. Add some more explanation to figure 5. As it is, it is difficult to understand (meaning of various colors). Add more details about the simulation setup. For example, the problems with stored data (in Figure 7a) are probably caused by some generated noise. In the paper its specification is completely missing. The used simulation tool should also be mentioned. The same holds for figure 12.

Specification

  • The article is rather descriptive.
  • I propose to shorten the article.
  • The text contains many unnecessary things (e.g. page 4 "Therefore, the authors said that the proposed local bit-line SRAM design effectively improves the read stability of the memory cells" ??).
  • Figure 5 (for example) - it is not explained what the orange / blue arrows mean (voltage? current?), what is the meaning of the diagram? unclear text (or I don't understand the text), also applies to other images.
  • Monte Carlo Figure 7, chapter 3.2 - simulation conditions, lin / worst case distribution, better description.
  • improve the overall processing of the article, description of images, context, meaning, comparison of simulations.

Author Response

The reviewer’s comments and suggestions are highly appreciated to improve the readability and quality of our manuscript. The manuscript has been revised and corrected typo, English mistake. The following point-to-point response for each comment given below along with the attached file.

Reviewer 2 Report

The authors presents 6T SRAM with local bitline scheme. The work is clearly presented and compared with previous works. Thus, I would recommend the paper to be published.

Author Response

(The authors gave the same response as above.)

Reviewer 3 Report

Authors propose a stable local bit-line 6T SRAM architecture for low-voltage operation, where RSMN, access time, bit line swing are improved and power leakages are decreased as compared with the conventional 6T SRAM architecture.

The proposed circuit is novel and interesting. However, the paper has serious flaws mainly due to poor English and organization. 

With this in mind, English proofreading and extensive editing of English language and style are strongly suggested. Moreover, the paper should be better re-organized. 

Some concerns and suggestions are listed in the following.

  • In the abstract, acronyms should be avoided. Moreover, the upperscript for the definition of the FoM has to be removed.
  • In introduction, references from 1 to 19 are not present, while the first cited is the 20th; references should be reported in sequence in the bibliography. A lot of typos are present and a lot of verb conjugations are wrong. Finally, a summary of the paper structure should be add.
  • Some achieved results are reported in the introduction, it is unnecessary. Authors can move results in the adequate position.
  • It is difficult to understand what is the proposed architecture, that on Fig. 2 or Fig.5 ? please, better clarify it.
  • In my opinion, section 2, 3 and 4 can be better organized without the use of short subsections.
  • Authors report many simulation results but PVT analisys is missed. Moreover, achieved results gathered in section 3 come from pre- or post-simulations? Please, specify it in the paper.
  • Since the layout is done, why there aren't measurements ? If the chip will be taped out soon, it would be better to insert measurements too. 
  • Table 3 has to not be splitted.

Author Response

(The authors gave the same response as above.)

Round 2

Reviewer 1 Report

This version of paper is much better.

Reviewer 3 Report

Authors have addressed my previous questions and made great improvement based on feedback from reviewers. I recommend a few more careful proofreading iterations to make sure the paper flows well.

One question is still not entirely satisfied, in particular:

  • PVT are not specified. I understand that reported results are for the worth case but what is this case? Please, specify the temperature of the worth case.